# WDR75: An essential protein for ribosome assembly undergoing purifying selection

**Lauren Lee, Justen B. Whittall** *

Department of Biology, Santa Clara University, Santa Clara, CA, United States of America

* jwhittall@scu.edu

**Data Availability Statement:** All relevant data are within the manuscript and its Supporting Information files.

**Funding:** The author(s) received no specific funding for this work.

## Abstract

Ribosomes, vital for life, consist of a large subunit and a small subunit (SSU), the latter is crucial for translation initiation and mRNA binding. The SSU processome, a 71-protein multimer in humans, is an intermediate in ribosome formation. One of its constituents, WDR75 plays a pivotal role by binding to an evolutionary conserved motif in the external transcribed spacer region of the rRNA to help form the SSU. Herein, we explore mammalian WDR75 molecular evolution, 3D structure, and phylogeny in light of its essential role in the SSU processome. We predict to find the footprint of purifying selection, especially at sites that are essential for proper ribosome assembly. In our comparison of 70 mammalian WDR75 sequences, we found ~25% of sites with significant purifying selection and no evidence of positive selection. Purifying selection was ~5x stronger for sites folding into beta-sheets than those predicted to be coils. Phylogenetic analysis validated expected mammalian relationships and uncovered an unusually long branch leading to mouse-eared bats, exhibiting 18x more substitutions per site than the average mammalian substitution rate. In testing for molecular evolution among branches, we found no evidence for purifying selection along any individual branches, but unexpectedly detected significant diversifying selection solely among African great apes.

## Introduction

Picture the ribosome as the molecular conductor of life's symphony, orchestrating the intricate dance of protein synthesis within every cell with seemingly humble components, the large and small subunits. The small subunit facilitates translation initiation by binding to mRNA, assisting in the positioning of tRNA, and ensuring accurate decoding of the genetic information for protein synthesis [1]. Ribosome biogenesis is a process in which more than 200 ribosome assembly factors catalyze the maturation of both ribosomal subunits. This cellular masterpiece, honed over 3–4 billion years of evolution [2]), not only shapes the very essence of life but also holds the keys to understanding ribosomopathies—human diseases caused by abnormalities in ribosome biogenesis, structure, or function [3]. Ribosomopathies arise from variants in either ribosomal proteins or the proteins responsible for creating the mature ribosome and can affect as many as 1 in every 50,000 live births [4]. For example, Diamond-Blackfan anemia is caused by multiple variants in certain ribosomal proteins themselves, alternatively, diseases like

**Competing interests:** The authors have declared that no competing interests exist.

Progressive Cholestasis of Northwestern Quebec (PCNQ) (formerly North American Indian childhood cirrhosis) are ribosomopathies associated with a variant in the UTP4 gene, a component of the SSU processome involved in small ribosomal subunit assembly. The ubiquity of ribosomes across the tree of life combined with their involvement in human diseases and multifaceted interaction networks suggest they should be largely evolutionarily conserved through strong purifying selection. Within this intricate landscape of ribosomal biology, the small subunit (SSU) processome stands out as a dynamic ensemble of ribonucleoprotein complexes, composed of at least 71 proteins in humans and 75 proteins in yeast [5]. It plays a critical role in early ribosomal maturation, guiding the assembly of the small ribosomal subunit [6].

One member of this SSU processome, U three protein 17 (UTP17), also known as the WD repeat domain 75 (WDR75), is a U3 small nucleolar RNA-associated protein. It is one of many SSU processome proteins that play essential roles in SSU ribosome biogenesis due to its interaction with the evolutionary conserved motif (ECM) within the pre-rRNA. WDR75 is a key component of the transcription-Utp complex (t-Utp), which is crucial for the transcription and early processing of pre-rRNA [7]. In particular, WDR75 functions in concert with two additional small nucleolar RNA-associated proteins in the SSU processome; UTP4 and UTP18 [8]. Furthermore, UTP4 and UTP18 associate with UTP6 and UTP10 to support assembly and stabilization of the tUTP complex whose interactions are important for ensuring the overall structure and function of the processome [9]. The specific quaternary protein-protein interactions between WDR75, UTP4, and UTP18 establish a binding site introduced by Singh et al. (2021) at amino acids 237–246, 288 and 321–322 in WDR75. The binding site recognizes an evolutionarily conserved motif (ECM) within the 5' external transcribed spacer (ETS) of the pre-rRNA for the SSU (Fig 1). This evolutionarily conserved region is unusual among regions of the rRNA that are removed in the final, mature 18S rRNA since these regions are typically more variable than the main 18S coding sequence (i.e. the internal transcribed spacer region

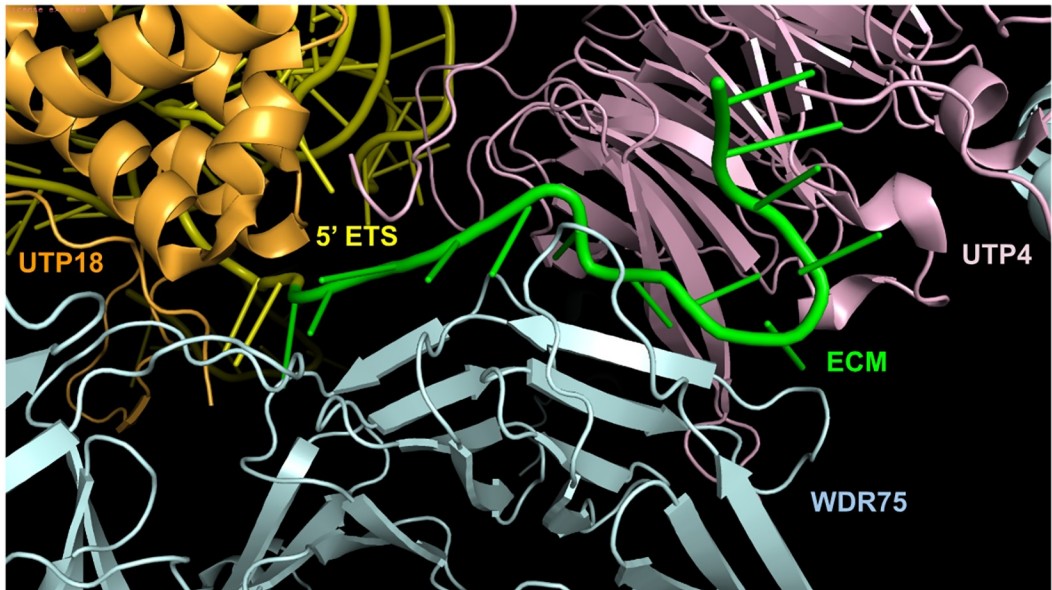

**Fig 1. The ECM of pre-rRNA interacts with WDR75, UTP4, and UTP18 proteins in the *Homo sapiens* SSU processome.** The ECM (green) is a portion of the ETS (yellow) which is part of the SSU pre-rRNA. The ECM interacts with WDR75 (blue), UTP4 (pink), and UTP18 (orange). This 3D structure (PDB: 7MQ8) was determined through Cryo-electron microscopy and visualized in PyMOL (Schrödinger, 2020).

in plants and fungi). However, the ECM has endured billions of years of evolution testament to its importance in the process of SSU ribosome biogenesis across Eukarya [10] suggesting both it and any interacting proteins will exhibit similar evolutionary histories (i.e. purifying selection on interacting protein coding genes like WDR75).As a member of the WD repeat-containing protein family, its primary structure consists mostly of 40–60 amino acid repeats and typically ends in its namesake peptides, tryptophan and aspartic acid (WD). The human WDR75 homologue has 13 WD repeats which secondarily fold into 57 beta strands. This secondary structure, especially the portion located on the outer surface of the SSU processome, which is well characterized, forms a propeller-like tertiary structure for the assembly of multi-protein complexes [11]. Thus, its structure, and the underlying amino acid sequence, are predicted to be under considerable evolutionary constraint because of its vast protein-protein network combined with its essential function in ribosome maturation [12].

Starting with the human WDR75, we travel back through evolutionary time by comparing across mammalian sequences [13]. In humans, WDR75 has multiple functions in ribosome development assembly including its essential role in pre-rRNA transcription and its role in cell survival via the p53 checkpoint. Though the mechanism by which WDR75 participates in RNA polymerase I transcription is unknown, the absence of WDR75 activates RPL5/RPL11-dependent p53 stabilization checkpoint [13]. More broadly, ribosomes, characterized by their complexity and intricate regulatory mechanisms, play a central role in cellular processes crucial for growth, division, and differentiation, however previous comparative studies have been limited. Extensive characterization of ribosome maturation has been conducted in yeast demonstrating widespread evolutionary conservation across Eukarya–at least the ~ 1 billion years that mammals and yeast have been separated [14].

By expanding our investigation of WDR75 across mammals, we can identify broader evolutionary patterns, identify any lineage-specific nuances and potentially contribute to the overall understanding of ribosome biogenesis and human ribosomopathies in particular. This study aims to uncover not only the molecular evolutionary history of this crucial protein in mammals but also its potential implications on ribosome biogenesis. To assess evolutionary conservation, we will examine evolution across phylogenetic branches and evaluate molecular evolution at individual amino acid sites making explicit comparisons between sites with known function and the remainder of the sites. Doing so could generate hypotheses for interactions for future functional characterization studies.

When testing for molecular evolution on the WDR75 coding sequence (CDS) across mammals, we are presented with a diversity of transcript variants. The production of transcript variants through alternative splicing is a pervasive phenomenon in eukaryotes and humans in particular. For example, in humans, over 70% of the ~25,000 characterized genes have transcript variants [15]. This process serves as a mechanism to diversify the proteome, allowing for a rich repertoire of proteins to emerge from a limited set of genes—humans average ~3.5 transcripts per gene [16]. Notably, transcript variants originating from the same gene can affect the primary, secondary and tertiary structures and thus display distinct functions, underscoring the complexity of alternative splicing and its relevance to protein function (protein-protein, protein-RNA interactions) in the case of WDR75 [17]. Some diseases are notably caused by alternative splicing. For example, approximately 13% of mutations in the CF transmembrane conductance regulator (CFTR) gene, which is associated with cystic fibrosis (CF), are categorized as "splicing mutations" [18]. Transcript variants may have different patterns of molecular evolution because they include/exclude different regions of a gene which may be under different levels of evolutionary constraint.

When comparing CDSs across species, differences in generation time may also account for variable rates of molecular evolution for the same gene [19]. The shortest-lived mammals,

such as small rodents like mice and shrews, may have generation times as short as a few months to a year. In contrast, the longest-lived mammals, such as some whales or elephants, can have generation times spanning several decades to over a century [20]. The varying generation times between small mammals and primates significantly influence the pace of molecular evolution, affecting the accumulation of genetic mutations and varying the molecular clock among lineages [21]. These differences will apply to most of the genome, in contrast to variable selective pressures on a particular gene or gene region. This prompts questions regarding the substitution rates in different mammalian subgroups, potential selective pressures on specific evolutionary branches, and the impacts of transcript variants on WDR75 structure.

This bioinformatics study employs molecular evolution, phylogenetic analysis, and 3D structural analyses to investigate the evolutionary history of mammalian WDR75. Our study aims to offer comprehensive insights into the evolutionary and functional nuances of WDR75 within the regulatory framework of mammalian cells. We explore selective pressures on WDR75 where it interacts with the rRNA ECM, UTP4, and UTP18, examine substitution rates among mammals and identify the impact of transcript variants on primary and secondary structure. Specifically, we address the following questions: 1) What selective pressures exist at interacting sites for WDR75? 2) Are there differences in selection among branches of the mammalian WDR75 tree? 3) What impacts do transcript variants have on the protein structure?

## Materials and methods

### Identifying sequences

Starting with the RefSeq for human WDR75 (NM_032168), we located homologous mammalian sequences from Genbank using blastn [22]. Cutoffs of 70% identity and E- value $< 1e-6$ were applied to increase the likelihood of locating homologous sequences [23]. Most sequences we used were from the RefSeq database, however additional human WDR75 sequences were included from the nucleotide collection (nr) using Megablast (S1 Table). For species with multiple transcript variants, the variant with the longest DNA sequence was selected except for *Homo sapiens* and *Pan paniscus* where all available transcript variants were included to determine the impact of alternative splicing on molecular evolution patterns, phylogenetic relatedness and 3D structures in these two taxa.

### Creating a multiple sequence alignment

Next, we created a DNA-based multiple sequence alignment (MSA) in order to identify patterns of sequence conservation, detect the footprints of selection, and conduct phylogenetic analyses. Sequences were imported into Geneious Prime version 2023.2.1 (https://www.geneious.com) as complete Genbank flatfiles. If a sequence did not already have a CDS annotation, the annotation was manually entered by translating the sequences and analyzing the three reading frames for a start and stop codon that aligned with the majority of the other sequences. With all sequences in the first reading frame, CDSs were extracted and aligned using Geneious' Translation Align algorithm with the BLOSUM90 cost matrix [24], gap open penalty from the default of 12 to14, and gap extension penalty from the default of 3 to 2. These parameters were generated through an iterative process of testing multiple parameter settings in order to maximize pairwise identity while minimizing gap insertions. Translation Align ensured gaps are inserted in multiples of three to maintain the reading frame. Since the transcript variants were substantially shorter than most of the other mammalian sequences, we investigated the effects of these truncated transcript variants on their secondary structure using PredictProtein [25].

## Phylogenetic analysis

Before generating phylogenetic trees, five additional outgroup sequences were added to the alignment from Genbank. To determine the most appropriate outgroup sequences to our study of WDR75 in mammals, we conducted a Discontiguous Megablast search with the *Homo sapiens* reference to the RefSeq database while excluding the taxon "Mammalia". We selected five reptiles (one alligator and four turtles) as outgroups. In Geneious, the outgroup CDSs were extracted and aligned to the mammalian MSA using Consensus Align with the same parameters described above (alignment files available in S2 Table).

We used three tree-building algorithms to generate a robust evolutionary framework for understanding the relationships among the studied DNA sequences and comparing the relative rates of substitutions among branches. For the distance analysis, we used the Geneious Tree Builder tool with the Jukes-Cantor distance model and the neighbor-joining tree-building method [26]. To assess the robustness of the inferred tree, bootstrap resampling was performed with 100 replicates. For phylogenetic analysis through parsimony, PAUP* software was launched from within Geneious using the maximum parsimony method and a heuristic tree search strategy. The outgroup was constrained to be monophyletic, and bootstrapping was executed with a random seed and 100 replicates to gauge the confidence among branches in the most parsimonious tree. Lastly, phylogenetic analysis through maximum likelihood was conducted using Randomized Axelerated Maximum Likelihood 8.2.11 (RAxML) [27] under the General Time Reversible model (GTR) with a discrete gamma distribution to estimate evolutionary distances categorizing substitutions into discrete categories of rates (CAT) while accounting for invariable sites (I). Rapid bootstrapping (100 replicates) and a search for the best-scoring maximum likelihood tree was carried out using the RAxML plug-in for Geneious.

## 3D structure analysis

The 3D structure of WDR75 was located by searching the PDB database using blastp of the human WDR75 protein accession (NP_115544). The PDB file 7MQ8 was a 100% match to the *Homo sapiens* protein and had an E- value of 0.0. This structure includes the small subunit processome consisting of 71 proteins in humans, one of which is WDR75 (chain LH). The 3D structure of the outer (surface) portion of WDR75 was visualized using PyMOL [28]. This 3D structure was originally determined using Cryo-EM by Sing et al. [11].

To locate interacting sites for WDR75 that we predict will show particularly strong footprints of purifying selection, we reviewed the relevant literature [11]. WDR75 interacts with UTP4 (chain LN) and UTP18 (chain LJ) to form a binding site for the ECM in the 5'-ETS of the pre-rRNA of the SSU (chain L0). To get a clear view of this site, chains LH, LJ, LN, and L0 were selected to make a new object while hiding the remainder of the SSU processome. Then, we examined the mammalian MSA at these interacting sites and recorded these residue's BLOSUM 62 values (an empirical predictor of the interchangeability of particular amino acids) and we noted the amino acid biochemical properties (charge, acidity, polarity, water solubility, etc.).

Unfortunately, there are no 3D structures available for the portion of the WDR75 embedded in the SSU processome. For these hidden residues with undetermined secondary and tertiary structures, we used PredictProtein (https://predictprotein.org/) [25] and Alpha Fold (https://alphafold.ebi.ac.uk/) [29, 30] to predict secondary structures (PredictProtein) and estimate a complete 3D structure (Alpha Fold) for *Homo sapiens* and *Pan paniscus* WDR75. To estimate the effects of transcript variants in *Homo sapiens* and *Pan paniscus*, we estimated their 3D structures using ColabFold v1.5.5: AlphaFold2 using MMseqs2 (https://colab.research.google.com/github/sokrypton/ColabFold/blob/main/AlphaFold2.ipynb) [31] and aligned them to the non-truncated Refseqs for these two species in PyMOL.

## Molecular evolution

**Single likelihood ancestor counting.** To examine patterns of molecular evolution for WDR75, the alignment was first prepared for Single-Likelihood Ancestor Counting (SLAC) [32] via the Datamonkey web portal [33–35] (https://www.datamonkey.org/slac). Outgroups and stop codons in the alignment were removed and alignment sites containing all gaps were stripped.

To test for purifying (negative) selection on a codon-by-codon basis, we estimated the rates of observed non-synonymous substitutions per non-synonymous site (dN) and synonymous substitutions per synonymous site (dS) using SLAC after choosing the universal genetic code. The resulting SLAC site table was sorted by the probability of purifying selection (P [dN / dS < 1]) and separately sorted by the intensity of purifying selection by sorting by (dN—dS) to determine sites exhibiting the most negative selection. Conversely, the descending order of the table when sorted by P [dN / dS > 1] and ascending dN-dS values were used to identify sites demonstrating the most diversifying (positive) selection. Groupings of sites based on their known secondary or tertiary structures (i.e., beta sheets vs. coils, known interacting sites vs. other sites, etc.) were tested for significance using the conservative non-parametric Mann-Whitney U-tests because of the non-normal distribution of dN/dS values and/or small samples sizes.

**Adaptive branch-site REL test for episodic diversification (aBSREL).** To test for episodic diversification along branches of our mammalian phylogenetic tree, we imported the same MSA used in the SLAC analysis into aBSREL [36] also using the Datamonkey web server (https://www.datamonkey.org/absrel). Therein, we selected all branches and the universal genetic code to test for episodic diversification measured as dN/dS values ($\omega$) within the phylogeny. A likelihood ratio test was applied to compare alternative models involving diversifying selection on some branches vs. models not including diversifying selection. The likelihood ratio test allows us to determine if the model with some branches undergoing diversifying selection is a significantly better fit to the data than the null model with no branches undergoing diversifying selection.

## Results

### Homologous sequence alignment

The final alignment consisted of 70 coding sequences of WDR75, 69 of which were Genbank RefSeqs (one additional, non-RefSeq sequence for *Homo sapiens* was included for comparison) (S1 Table; alignment files available in S2 Table). The global nucleotide pairwise identity for these CDSs was 89.9% (max = 99.960%, min = 73.181%). The mean sequence length of the CDS was 2,487 bp (SEM ±32.4 bp) (829 amino acids). Within the first 1,838 bp of the *Homo sapiens* CDS there are 13 WD repeats folding into 57 beta strands with each repeat containing approximately 40 amino acids (Fig 2). For the transcript variants for *Homo sapiens* and *Pan paniscus*, both are missing the first 193 bp of the coding sequence which includes the first and approximately half of the second WD-repeat (according to the reference sequence annotation NM_032168). According to PredictProtein, this deleted region includes six strands which are predicted to fold into beta-sheets.

### Variable branch lengths and the footprint of selection along phylogenetic branches

Phylogenetic analyses from three algorithms produced comparable trees that were largely congruent with accepted mammalian relationships with no strong (bootstrap >70%) conflict, yet

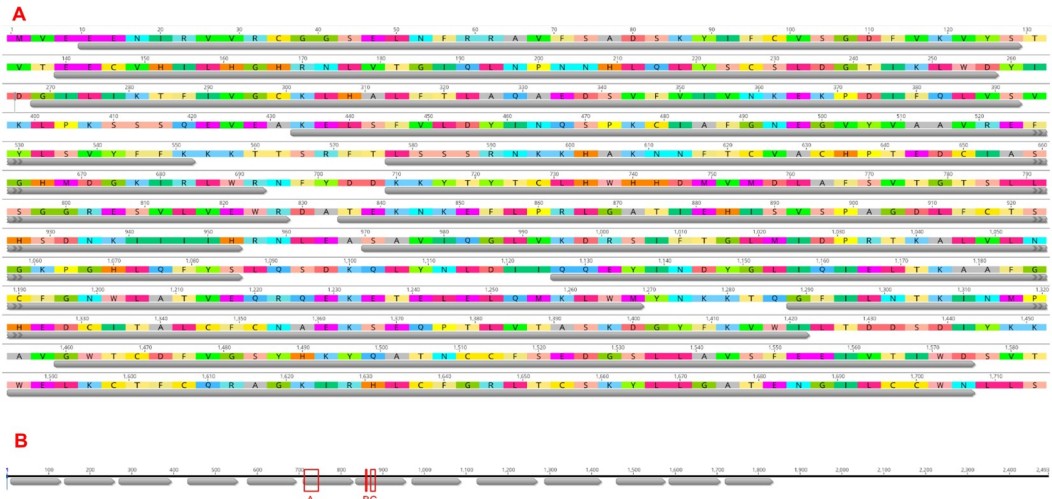

**Fig 2. WD Repeats found in the WDR75 reference sequence:** *Homo sapiens* **(NM_032168).** (A) Translation of the first 1845 out of 2493 bp are shown for the *Homo sapiens* sequence. WD repeats are shown with grey annotations spanning the repeat. The 13 WD repeats in WDR75 are annotated with grey arrows with each repeat containing approximately 40 amino acids. A summary diagram of the entire CDS and the relative location and size of the WD repeats for the humans shown in (B). Red boxes in (B) indicate known interacting sites based on the literature (Singh et al. 2021).

there were some weakly supported/unresolved branches as expected for mammals using a single locus (Fig 3; S1 & S2 Figs). Although the topologies were very similar to expected relationships, there were some unexpectedly long branches recovered in all three phylogenetic analysis methods. First, the stem branch leading to Myomorpha (mouse-like rodents) has 7.78 times more substitutions per site than the rest of Rodentia (Table 1). Another branch leading to *Microtus oregoni* (creeping vole) has 8.51 times more substitutions per site than Rodentia (Table 1). Additionally, the stem branch leading to the genus *Myotis* (mouse-eared bats) has 114.43 times more substitutions per site than the stem branch leading to the rest of Chiroptera (bats) and 18x more substitutions per site than the mammalian average (Table 1). Diversifying selection emerged solely along the Human-Bonobo transcript variant branch with an exceptionally high dN/dS value reflecting a near zero dS value in the denominator (aBSREL, likelihood ratio test p < 0.05, Fig 3).

## 3D structure analysis

Amino acids 236–247 in human WDR75 interacts with UTP4 and the ECM forming beta sheets and coils (Fig 4) and is located in the sixth WD repeat (Fig 2B).

To investigate the effects of transcript variants on the structure of WDR75, we examined the primary structure (amino acids) in Geneious, then utilized PredictProtein to determine the effects on secondary structure and AlphaFold to predict how the 3D structures of these variants, which contain late start codons, differ from the existing 3D model of WDR75 (PDB: 7MQ8). At the primary structure scale, 64 amino acids are missing from the coding sequence of these two transcript variants in both human and bonobo compared to the rest of the mammalian MSA. Additionally, the first seven amino acids in these variants differ from the remainder of the alignment, which could impact their secondary and tertiary structures. As a result, both the human and bonobo transcript variant sequences fold into 54 beta strands, which is three fewer than the human and bonobo reference sequence (Fig 5A, 5C, 5F and 5G).

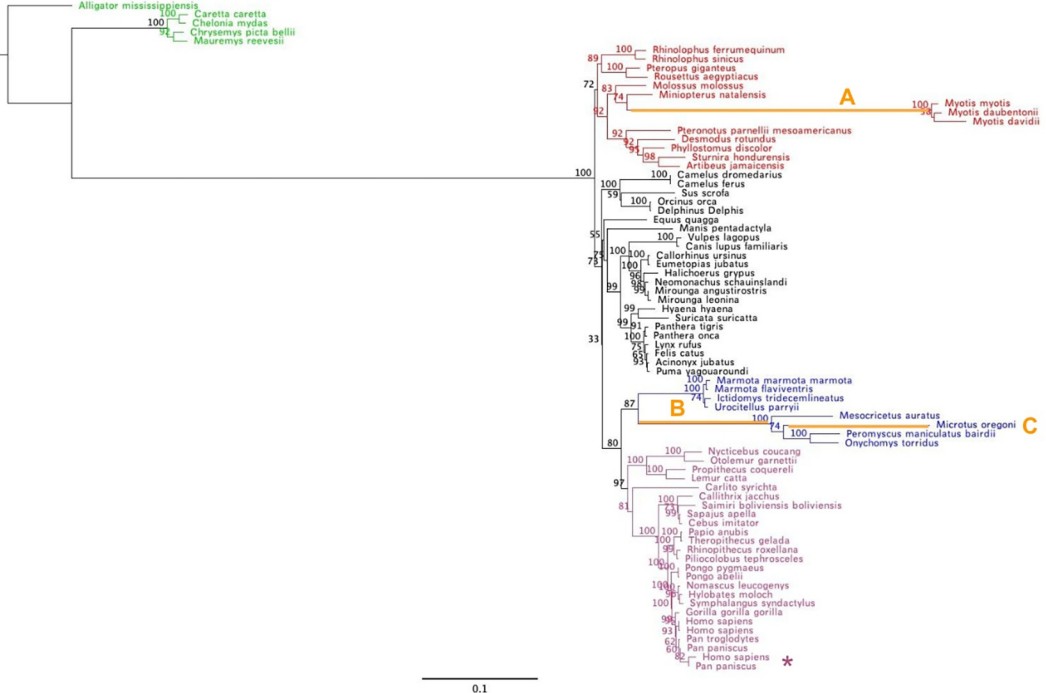

**Fig 3. Maximum likelihood phylogram of mammalian WDR75 based on DNA sequences.** This RAxML tree of mammals is rooted with reptiles, contains bootstrap values along the branches which are shown proportional to their inferred number of substitutions (subs/site). Green branches identify the reptile outgroups rooted with *Alligator mississippiensis*. Chiroptera (red), Rodentia (blue), and primates (purple) are identified as colored branches and with tip labels. The remaining mammals, mostly Carnivora and Artiodactyla, are shown in black. The three exceptionally long branches have been labeled (A) *Myotis* (bats), (B) the rodent suborder Myomorpha, and (C) the rodent *Microtus oregoni* (creeping vole). The asterisk identifies the transcript variants for *Homo sapiens* and *Pan paniscus*. The scale bar is in units of substitutions per site.

## Molecular evolution

For the UTP4 and ECM binding sites of the *Homo sapiens* WDR75 CDS, the dN-dS value for beta-sheet amino acid codons exhibit 5.3 times stronger negative selection than amino acids located in coils (SLAC analysis, Mann-Whitney U-test, z-score = 2.78571, p = 0.0053, Table 2, S3 Table). The average dN-dS value for significant purifying interacting sites is 2.14 times more purifying than the average dN-dS value for the entire WDR75 CDS (Mann-Whitney U-test, p < 0.05, S3 Table). In the *Homo sapiens* WDR75 CDS, out of 837 total amino acids, there are 319 sites exhibiting negative selection (38.11%) and no evidence for significant diversifying

**Table 1. Clade support and stem branch lengths from the maximum likelihood phylogram of mammals' *WDR75* coding sequences.**

| Branch | Bootstrap Value | # Substitutions per site |
|---|---|---|
| **Primates** | 97 | 0.0054 |
| **Rodentia** (rodents) | 87 | 0.0148 |
| **Myomorpha** (mouse-like rodents) | 100 | 0.1151 |
| ***Microtus oregoni*** (creeping vole) | NA[1] | 0.1260 |
| **Chiroptera** (bats) | 72 | 0.0023 |
| ***Myotis*** (mouse-eared bats) | 100 | 0.2632 |

[1]No bootstrap value since this is not a clade, but instead just a tip of the tree.

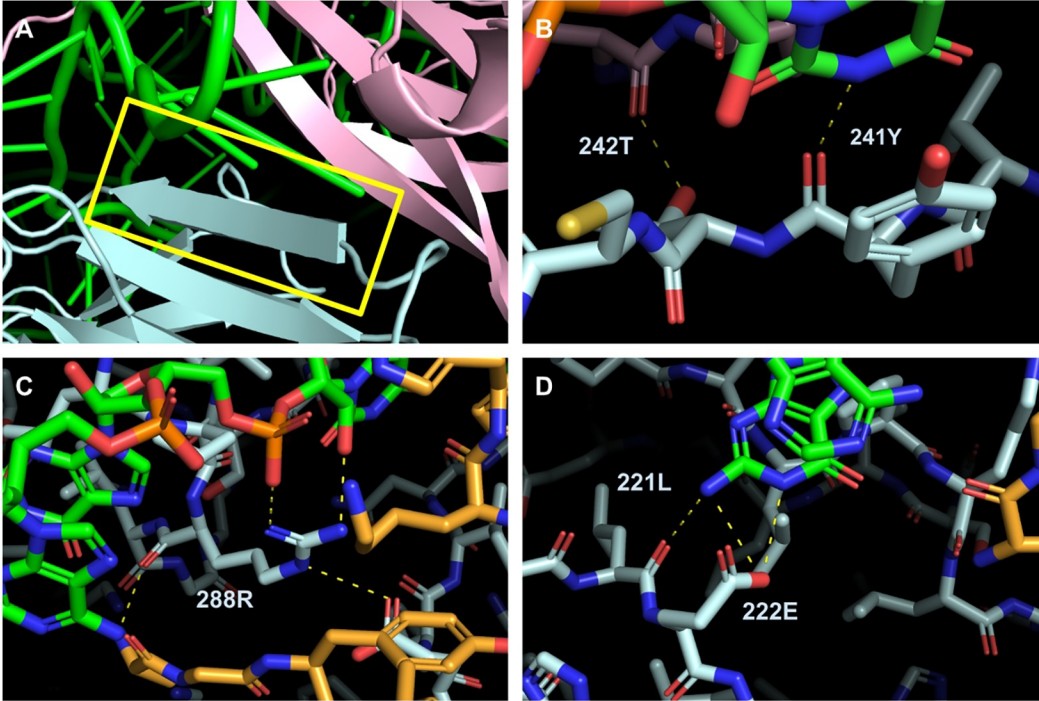

**Fig 4.** *Homo sapiens* **WDR75 interactions within 4 Å of the ECM, UTP18 and UTP4.** The image of the PDB 7MQ8 is taken using PyMOL and displays cartoon WDR75 in blue, UTP4 in pink, UTP18 in orange, and the ECM in green. The remaining, non-interacting proteins present in the SSU processome multimer are hidden for simplicity. In panel A, WDR75 amino acids interacting with ECM and UTP4 are boxed in yellow and are located from amino acid sites 237 to 246. Beta sheets are shown as arrows. Panel B shows polar interactions (yellow dashed line) between WDR75 and UTP4 (pink) and ECM (green). Panel C show polar interactions between WDR75, UTP18, and ECM while panel D shows polar interactions between WDR75, and the ECM.

(positive) selection (SLAC analysis, all $p < 0.05$ are for negative dN-dS values, Table 2, S3 Table).

## Discussion

This study examined the evolutionary history of mammalian WDR75, a key protein involved in ribosome biogenesis, by investigating the footprint of selection on amino acid sites in light of their secondary/tertiary structures and along branches of the mammalian phylogeny. Evolutionary conservation at the amino acid scale appears to be the rule for the majority of the WDR75 CDS, assumedly to maintain the integrity and functionality of the protein during ribosomal small subunit maturation. Thoroughly analyzing the 3D structure in PyMOL, we visualized the known interacting sites with UTP4 and ECM, introduced by Singh et al. [11], on WDR75 as amino acid sites 237–246, 288 and 321–322 in our alignment (Fig 4, S5 Fig). WDR75 is predominantly governed by negative (purifying) selection at the ECM binding site formed by WDR75, UTP4, and UTP18. We found no evidence for significant diversifying selection on any amino acid sites across the coding sequence. This finding underscores the importance of conserving the function of WDR75 in ribosome biogenesis across diverse mammalian species. While maintaining overall protein structure contributes to sequence conservation of many proteins including WDR75, its interaction with the rRNA's ECM in the developing SSU is likely an important factor in sequence conservation. A previous study on mammalian UTP6 (one of the many proteins in the SSU processome that does not interact

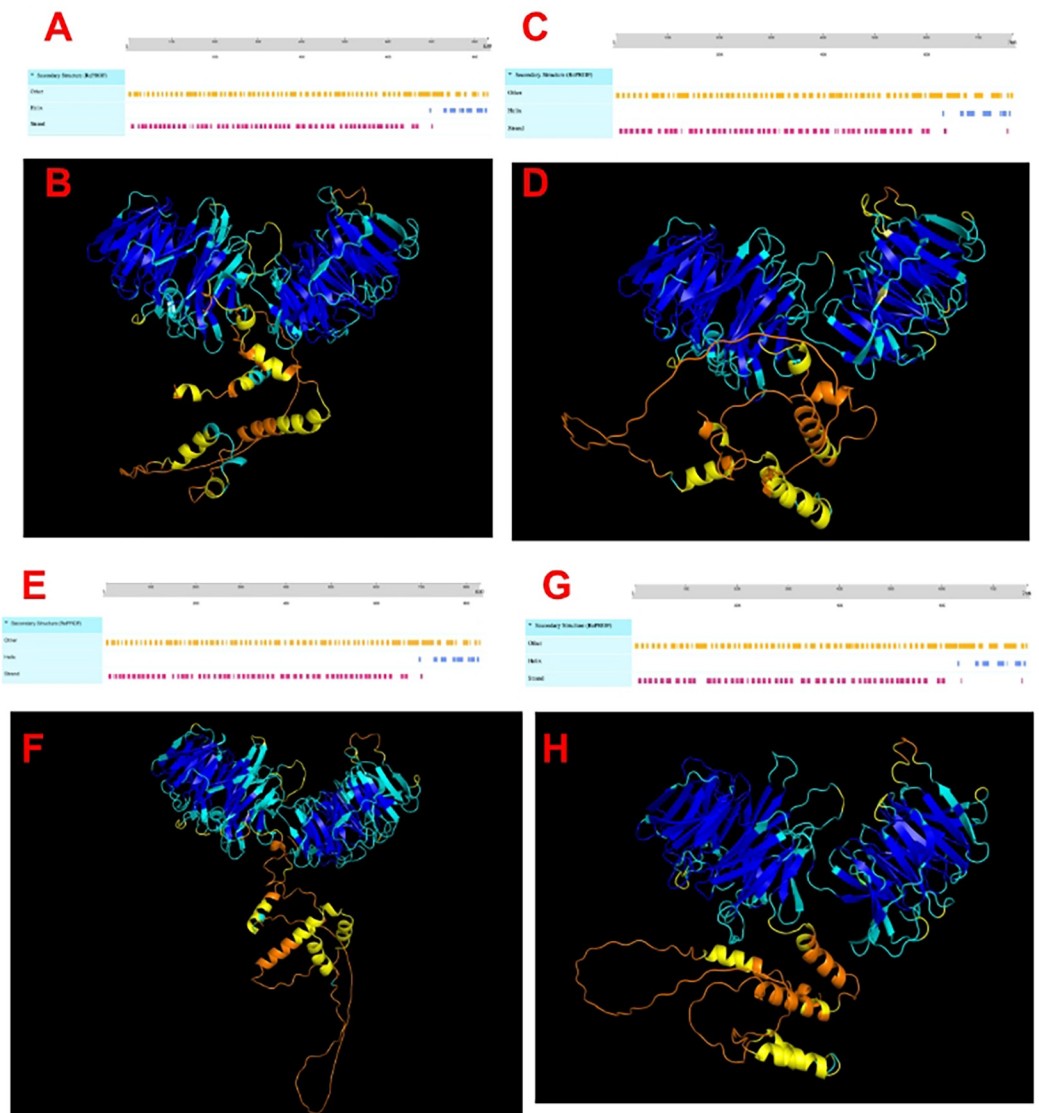

**Fig 5. Predicted secondary and 3D structures of human and bonobo WDR75 and their WDR75 transcript variants.**
Predicted secondary structure of the reference human WDR75 sequence in panel A with predicted 3D structure in panel B
(Q8IWA0). Predicted secondary structure of the human WDR75 transcript variant in panel C with predicted 3D structure in
panel D (d1143). Predicted secondary structure of the bonobo WDR75 reference sequence in panel E with predicted 3D
structure in panel F (A0A2R8ZEG9). Predicted secondary structure of the bonobo WDR75 transcript variant in panel G with
predicted 3D structure in panel H (5e91a). Colors in the 3D structures represent model confidence (pLDDT out of 100) with
dark blue as very high (pLDDT > 90), light blue as high (90 > pLDDT > 70), yellow as low (70> pLDDT> 50) and orange
as very low (pLDDT< 50) confidence.

with the ECM) reports evidence of diversifying (positive) selection at 15 amino acid sites
across 12 mammalian sequences ([37] & their S5 Table). The conservation of the WDR75
interacting sites suggests a footprint of purifying selection maintaining the integrity and effi-
ciency of the ribosome assembly process- particularly the binding of pre-rRNA in the SSU pro-
cessome [11]. Although depletion of most SSU processome proteins in yeast results in a pre-
rRNA processing defect (i.e. they still produce the 35S), however for the members of the
UTP-A/t-Utp subcomplex (which includes UTP17/WDR75), their depletion causes an addi-
tional reduction in pre-rRNA transcription (i.e. diminished production of the 35S). Our

**Table 2. Evidence of selection on WDR75 of *Homo sapiens* and related mammals at known interacting sites with UTP4, UTP18, and the ECM.**

| Amino Acid | Biochemical property | Interaction | Secondary structure | dN-dS value |
| --- | --- | --- | --- | --- |
| 237K | Positive Charge | UTP4 | Coil | -0.955 |
| 238K | Positive Charge | UTP4 | Coil | 0.296 |
| 239Y | Hydrophobic | pre-rRNA ECM | Coil | 0.336 |
| 240T | Polar uncharged | UTP4 | Beta Sheet | -2.595 |
| 241Y | Hydrophobic | pre-rRNA ECM | Beta Sheet | -4.999* |
| 242T | Polar uncharged | UTP4 | Beta Sheet | -3.179* |
| 243C | Contains Sulfur | pre-rRNA ECM | Beta Sheet | -5.014* |
| 244L | Hydrophobic | UTP4 | Beta Sheet | -4.210* |
| 245H | Positive Charge | pre-rRNA ECM | Coil | -1.571 |
| 246W | Hydrophobic | pre-rRNA ECM | Coil | -1.895 |
| 288R | Positive Charge | pre-rRNA ECM and UTP18 | Coil | -2.621* |
| 321L | Hydrophobic | pre-rRNA ECM | Coil | -1.934* |
| 322E | Negative Charge | pre-rRNA ECM | Beta Sheet | -4.056* |

*p-value < 0.05 indicates significant evidence of negative selection at that amino acid site across the alignment.

primarily negative selection results align with the essential nature of WDR75 in not only the processing, but also maintaining the transcription of rRNA during ribosome biogenesis [8]. This emphasizes its evolutionary significance in cellular processes fundamental for life, and gives insights into the selective pressures on WDR75's unique role originally reported by Singh et al. [11].

In the MSA, the start codons for all CDSs appears within 1–2 amino acids of each other except the transcript variants (X2) from *Homo sapiens* and *Pan paniscus* (bonobo)—the only transcript variants included in this MSA (Fig 6). In addition to the late start codon, the first seven amino acids disagree with the rest of the alignment (Fig 6), prompting an investigation to identify selective pressures acting on specific branches of the phylogenetic tree. Diversifying selection detected exclusively on the transcript variant branches suggests potential adaptive evolution or functional divergence associated with transcript variants of WDR75 in these closely related species, however the exceptional dN/dS value should be interpreted with caution due to a near zero dS value in the denominator. Further structural analyses are warranted to elucidate the precise effects of these transcript variants on WDR75 protein structure and ribosome biogenesis. We predict that in the case of WDR75, the investigated transcript variants will have considerably lower expression compared to their respective reference sequences because of the effects the transcript variant has on secondary and predicted 3D structure (Fig 5 and S3 Fig), however this remains to be tested. Future studies examining the expression levels across tissues of transcript variants will be necessary to understand the functional implications of the structural changes associated with these transcript variants.

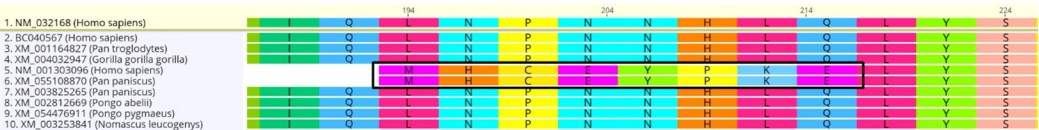

**Fig 6. *Homo sapiens* and *Pan paniscus* transcript variants X2 start at base pair 193 in the reference WDR75 CDS.** Shown above is the ingroup multiple sequence alignment of the WDR75 coding sequence in Geneious using translation align. *Homo sapiens* and *Pan paniscus* transcript variant X2 CDSs are indicated with red stars. The black boxes show the first seven amino acids in the transcript variant sequence.

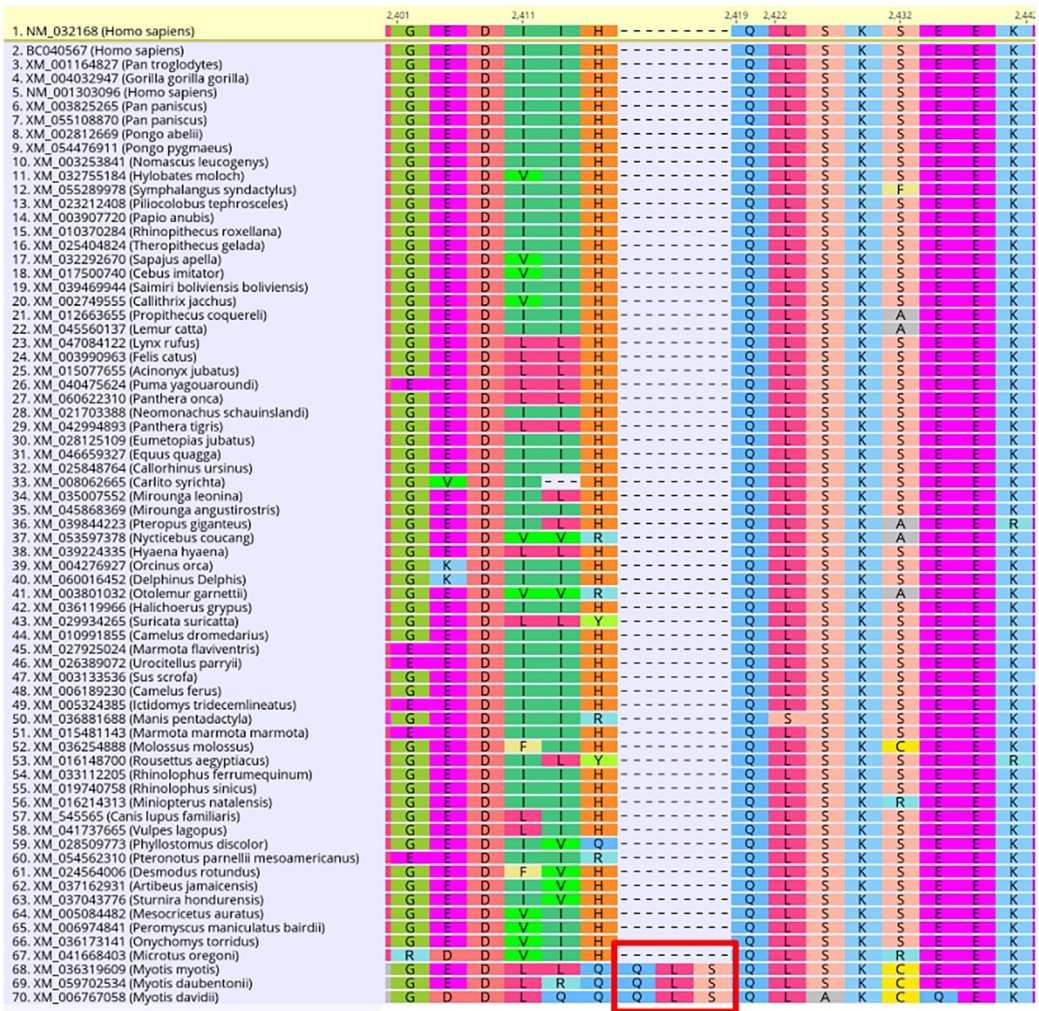

**Fig 7. Amino acid insertion in the *Myotis* WDR75 coding sequence at base pair 2419 of the reference sequence.** Shown above is the ingroup multiple sequence alignment of the WDR75 coding sequence in Geneious using translation align. The red box indicates the insertion of 3 amino acids in the *Myotis* sequences: *Myotis myotis*, *Myotis daubentonii*, *Myotis davidii*. The sequences in the alignment are sorted by the # of differences from the reference sequence, *Homo sapiens*, which is highlighted in yellow.

Most branches in the RAxML tree have tips that terminate at approximately the same distance from the root suggesting a relatively constant molecular clock. However, *Microtus oregoni* (creeping vole) exhibits a 7–9 times higher substitution rate (#substitutions per site) than the rest of Rodentia, indicating potentially unique evolutionary dynamics in this species, unique to this gene, or the intersection between the two (Fig 3). Two lineages had exceptionally long branches, including the genus *Myotis* (mouse-eared bats) and Myomorpha (mouse-like rodents) indicating elevated substitution rates (Fig 3). *Myotis* has an insertion of three amino acids at site 2419 compared to the remaining sequences in the MSA (Fig 7), but when these sites were removed, the long branch persists (S4 Fig). Overall, these exceptionally long branches are consistent with the current literature on mammalian molecular evolution that shorter generation times are correlated with higher substitution rates across their genomes. Rodents, like mice, have a generation time of several months compared to decades in primates

and a 2-3-fold higher substitution rate per year [38]. The elevated substitution rates observed in *Myotis* and Myomorpha are most likely attributed to their shorter generation times, which are correlated with higher genomic substitution rates across their genome, including WDR75/Utp17.

### Limitations and future directions

While this study offers insights into the evolutionary and functional nuances of WDR75 in mammalian ribosome biogenesis, it is not without limitations. 3D structure analysis was limited to existing PDB files of human SSU processomes in which WDR75 is a single chain (out of 71 in humans), existing files were not able to model many of the residues buried in the SSU processome structure including several alpha helices and coils comprising the C-terminus. When predicting the 3D structure of WDR75 using AlphaFold, the alpha helices and coils between helices missing from the PDB were modelled with low (70> pLDDT> 50) to very low confidence (pLDDT< 50). Because of the low confidence, further studies on WDR75 should consider experimental methods to determine the remainder of the monomeric 3D structure.

Overall, this study contributes to the growing body of knowledge on WDR75's role in ribosome biogenesis and provides a small contribution to our understanding of the molecular evolution of proteins involved in cellular processes essential for life.

### Supporting information

**S1 Table. Homologous WDR75 coding sequences used in this study based on the *Homo sapiens* reference (NM_032168).**
(DOCX)

**S2 Table. Mammalian WDR75 nucleotide and protein alignment files.**
(DOCX)

**S3 Table. Amino acid sites exhibiting significant purifying selection using Datamonkey's SLAC online server for the entire WDR75 coding sequence across mammals.**
(DOCX)

**S1 Fig. Distance-based phylogenetic analysis of WDR75 alignment for mammals.** This neighbor- joining tree shows results from 100 bootstrap replicates—indicated along the branches. Green branches identify the reptile outgroups rooted with *Alligator mississippiensis*. Chiroptera (red), Rodentia (blue), and primates (purple) are colored branches and tip labels. The remaining mammals, mostly Carnivora and Artiodactyla, are shown in black.
(DOCX)

**S2 Fig. Parsimony-based phylogenetic analysis of WDR75 alignment for mammals.** Green branches identify the reptile outgroups rooted with *Alligator mississippiensis*. Chiroptera (red), Rodentia (blue), and primates (purple) are colored branches and tip labels. The remaining mammals, mostly Carnivora and Artiodactyla, are shown in black.
(DOCX)

**S3 Fig. Alpha-fold predicted 3D structures of human and bonobo WDR75 aligned with their transcript variants.** Panel A shows the predicted human WDR75 3D structures and panel B shows the predicted bonobo WDR75 3D structures. Blue structures are the complete reference sequences while yellow structures indicate the transcript variants that are shortened on the N-terminus.
(DOCX)

**S4 Fig. Maximum likelihood phylogenetic analysis of WDR75 alignment for mammals with the additional three codons in *Myotis* removed (at reference sequence DNA site 2419).** Green branches identify the reptile outgroups rooted with *Alligator mississippiensis.* Chiroptera (red), Rodentia (blue), and primates (purple) are colored branches and tip labels. The remaining mammals, mostly Carnivora and Artiodactyla, are shown in black. The long branches are still present even though the inserted codons in *Myotis* were removed.
(DOCX)

**S5 Fig. Mammalian WDR75 amino acid multiple sequence alignment showing high residue conservation at interacting sites.** *Homo sapiens* RefSeq (NM_032168) is the reference in yellow at the top. Dots indicate identical amino acids to the reference. Box A shows amino acid sites 237–246 (bp 709–728), B shows amino acid site 288 (bp 862–864), and C contains amino acids 321 and 322 (bp 961–966) (see Fig 2 for alignment wide location of A, B, C). Note: there is a break in the alignment between regions A and B+C.
(DOCX)

## Author Contributions

**Conceptualization:** Lauren Lee, Justen B. Whittall.

**Data curation:** Lauren Lee, Justen B. Whittall.

**Formal analysis:** Lauren Lee, Justen B. Whittall.

**Investigation:** Lauren Lee, Justen B. Whittall.

**Methodology:** Lauren Lee, Justen B. Whittall.

**Project administration:** Lauren Lee, Justen B. Whittall.

**Resources:** Lauren Lee, Justen B. Whittall.

**Software:** Lauren Lee, Justen B. Whittall.

**Supervision:** Justen B. Whittall.

**Validation:** Lauren Lee, Justen B. Whittall.

**Visualization:** Lauren Lee, Justen B. Whittall.

**Writing – original draft:** Lauren Lee.

**Writing – review & editing:** Lauren Lee, Justen B. Whittall.

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
