## [Decision Letter · Decision Letter 0]

17 Sep 2024

PONE-D-24-35385WDR75: An essential protein for ribosome assembly undergoing purifying selectionPLOS ONE

Dear Dr. Whittall,

Thank you for submitting your manuscript to PLOS ONE. After careful consideration, we feel that it has merit but does not fully meet PLOS ONE’s publication criteria as it currently stands. Therefore, we invite you to submit a revised version of the manuscript that addresses the points raised during the review process.

We look forward to receiving your revised manuscript.

Kind regards,

Jorge Perez-Fernandez, Ph.D.

Academic Editor

PLOS ONE

**Journal Requirements:**

Reviewers' comments:

Reviewer's Responses to Questions

**Comments to the Author**

1. Is the manuscript technically sound, and do the data support the conclusions?

Reviewer #1: Partly

Reviewer #2: Yes

2. Has the statistical analysis been performed appropriately and rigorously? 

Reviewer #1: N/A

Reviewer #2: N/A

3. Have the authors made all data underlying the findings in their manuscript fully available?

Reviewer #1: Yes

Reviewer #2: No

4. Is the manuscript presented in an intelligible fashion and written in standard English?

Reviewer #1: Yes

Reviewer #2: Yes

5. Review Comments to the Author

**Reviewer #1: **In this ms from Justen Whittall's lab, the authors examine the conservation of the SSU processome protein WDR75 (Utp17) in mammals.

The manuscript is interesting from the perspective that it examines the conservation of this ribosome assembly protein - most of this type of work has been functional and little has been of an evolutionary nature, thus the interest of this paper.

However, there are a number of serious problems with this paper.

Many of the problems I have directly marked on the paper and uploaded a marked up PDF copy of the paper to the review portal.

The main problems namely are:

1-The authors consistently call the SSU processome a 71-protein multimer. It is not clear till later in the paper as to why they use the number 71 - which could be the number of proteins in the human SSU processome cryoEM paper. In yeast, there might be as many as 75 proteins in the SSU processome, though some of these are yeast-specific proteins. Regardless, other than a mention that there are 71 to 75 proteins in the SSU processome, the complex should be called by its proper name, which is the SSU processome not the 71-protein multimer.

2-In the abstract, line 21, it is incorrect to call WDR75 a ribosomal protein. It is not a ribosomal protein, but rather an SSU processome protein.

3-Many of the references in the ms need to be updated with more recent references.

4-Lines 49/50-While Diamond-Blackfan anemia is likely the most common ribosomopathy, North American Indian Childhood cirrhosis is not common and is in fact quite rare. For that reason, I would remove it from that sentence. The authors might choose to keep it though because their protein of interest, WDR75/Utp17 is a member of the UTP-A/t-Utp subcomplex (a fundamental concept that should be included in this paper but is not) and North American Indian Childhood cirrhosis is a ribosomopathy affecting a different member of the Utp-A/t-Utp subcomplex. This disorder should aslo be referred to by its newer name, Progressive cholestasis of northwestern Quebec (PCNQ). It should also be noted that while Diamond-Blackfan anemia is in a ribosomal protein, NAIC/PCNQ is in an SSU processome protein. Lastly, human diseases should not be called mutations but rather variants.

5-Throughout the paper, there is mention of the importance of WDR43/Utp17. It is not clear (an argument is not made) for why Utp17 is important. Utp17, like most SSU processome proteins, is highly conserved and is an essential protein (as are about 86% of SSU processome proteins). What makes WDR43/Utp17 more important than the other highly conserved and essential SSU processome proteins?

6-In many places in the ms, there is mention of the 57 beta sheets of WDR43/Utp17. Do the authors mean 57 beta sheets (seems high) or rather 57 beta strands?

7-The naming convention for the Utp proteins is incorrect: it is U three protein.

8-In line 76, I do not agree with the statement that the ITSs are hypervariable. They are more variable than the main 18S coding sequence, but I would not call them hypervariable.

9-All of the images in the ms are of very low resolution and difficult to read.

10-In lines 96 to 102, there appears to be confusion (as far as I can tell) about the Budkevich reference. A quick review of that reference suggests that it looks at structural rearrangements during translation, not during ribosome assembly. It does so in rabbit but not in cow (contrary to the text). It appears that the use of this reference is out of place and possibly confusing ribosome assembly and translation. Further, the comparison to ribosome assembly in bacteria is maybe not useful as we know that bacterial ribosome assembly is completely different from the process in eukaryotes (bacteria don't have an SSU processome). Lastly, in lines 100-102, this statement is misleading. First, a characterization of ribosome assembly in humans vs yeast does not qualify as widespread evolutionary conservation as yeast and humans are quite closely related. There has been characterization of ribosome assembly in truly distant eukaryotes such as Euglena, trypanosome, and Giardia where that comment would be true of widespread evolutionary conservation.

11-Line 106 talks about an aim of the study being to uncover the molecular evolutionary history of WDR75/Utp17. It is unclear to me that that can be achieved by only looking at the sequence is mammals. In order to achieve that goals, the sequence of Utp17 would need to be examined throughout Eukaryotes.

12-Line 121-while I agree that alternative splicing is important and that it has overall been understudied in ribosome assembly, it is unclear how the study of alternative splicing in WDR75/Utp17 is relevant to ribosomopathies considering that there are no known (as far as I know) ribosomopathies in WDR75/Utp17.

13-In the materials and methods, it is unclear if the multiple sequence alignments/trees where built from DNA or protein sequences. The implication is that they were built from DNA sequences. If so, why DNA and why not protein sequences? Also, where the alignments manually edited (they should be) after having been generated?

14-In the results section of page 12, why not give the results of the protein alignment? Wouldn't that be more informative that the results of the nucleotide alignment? Also, the protein alignment should be included as a supplemental figure.

15-In Figure 5, why not include the bonobo reference sequence and structure in order to do a better comparison between the full-length human and bonobo sequence/structure?

16-In the section on molecular evolution/sequence conservation, how do the authors know that these sequences are conserved due to their particular function (protein-protein, protein-RNA interactions) and not simply conserved due to maintaining the overall structure of WDR75/Utp17?

17-Table 2 might be more useful if it included the function of the particular amino acids, such as interacting with another protein (in which case, which one?) or RNA?

18-A protein alignment of WDR75/Utp17 would be helpful. the region identified by the authors (aa 237 to 322), how conserved is it relative to the rest of the protein? An alignment would make this clear.

19-Lines 367-370 is partially incorrect. Depletion of most SSU processome proteins in yeast gives a pre-rRNA processing defect (as in, pre-rRNA transcription is normal (production of the 35S) but there is a block in the pre-rRNA processing/cleavage. However, for the members of the UTP-A/t-Utp subcomplex, their depletion results in both a decrease in pre-rRNA transcription (loss of the 35S) AND pre-rRNA processing.

20-Line 373-374: are there other alternatively spliced isoforms of WDR75/Utp17? or only the one mentioned with the downstream start codon?

21-Lines 383-384 talks about WDR75/Utp17 transcript variants being low expression. What is this based on?

22-The bottom paragraph of page 20 talks about faster substitution rates. Are these faster substitution rates know for those particular organisms, beyond the comment of their short reproductive life cycle? This makes it difficult to determine if WDR75/Utp17 has a high substitution rate in those organisms or if that is a feature of all genes in those organisms.

23-Line 418 and elsewhere talks about there being only a single chain of WDR75 in the human SSU processome cryoEM structure. This is likely because WDR75/Utp17 is a single-copy protein within the SSU processome. It is unclear why this statement is made here. Why is this being said?

**Reviewer #2: **The manuscript provides a comprehensive analysis of the conservation and molecular evolution of the WDR75 protein. The authors conducted a multiple sequence alignment across mammalian species, inferred a phylogenetic tree, and integrated this with both experimentally determined and predicted 3D structures. They report evidence of purifying selection and identify an unusually high substitution rate in bat lineages, attributing this to short generation times. Overall, the manuscript is well written, the approach is sound, and the results could be of interest to others studying this protein.

Major comments

The analysis depends heavily on the Geneious software, which is a paid, proprietary platform. This poses potential limitations for reproducibility, as not all readers may have access to this software. Could the authors elaborate on their choice of Geneious over open-source tools? Was there any specific functionality or feature in Geneious that could not be replicated using freely available software?

Minor comments

Given that multiple sequence alignment is central to the analysis, I recommend making the complete MSA available in Stockholm format in the supplementary materials. This would allow others to scrutinize the alignments more easily and potentially reuse them for future studies.

Similarly, since the manuscript involves predicted structures, I suggest including the predicted 3D models in widely accessible formats, such as PDB or mmCIF, as supplementary files. This would be beneficial for reproducibility, especially given that prediction algorithms (such as those used by AlphaFold) may improve or change over time, potentially affecting the reproducibility of the structural analysis.

Figures 2, 3, 5, 6, and 7 were unreadable in the provided PDF proof. Please ensure that all figures are presented with high resolution and are properly embedded in the final submission.

Typos and Style Corrections

1. In the sentence “While my study offers…”, the wording should be revised to “While this study offers…”.

2. In the sentence “maturation in ‘mammals’ and even ‘eukaryotes’ in comparison to bacteria,” the use of quotation marks around mammals and eukaryotes is unnecessary and may create confusion.

6. PLOS authors have the option to publish the peer review history of their article (what does this mean?). If published, this will include your full peer review and any attached files.

Reviewer #1: No

Reviewer #2: No

---

## [Decision Letter · Decision Letter 1]

19 Nov 2024

PONE-D-24-35385R1WDR75: An essential protein for ribosome assembly undergoing purifying selectionPLOS ONE

Dear Dr. Whittall,

Thank you for submitting your manuscript to PLOS ONE. After careful consideration, we feel that it has merit but does not fully meet PLOS ONE’s publication criteria as it currently stands. Therefore, we invite you to submit a revised version of the manuscript that addresses the points raised during the review process.

As noted by Reviewer 1, there are several areas in the manuscript that require further clarification or revision. Please address these concerns directly, providing a rationale for any decisions to retain the original text.

We look forward to receiving your revised manuscript.

Kind regards,

Jorge Perez-Fernandez, Ph.D.

Academic Editor

PLOS ONE

Journal Requirements:

**Comments to the Author**

1. If the authors have adequately addressed your comments raised in a previous round of review and you feel that this manuscript is now acceptable for publication, you may indicate that here to bypass the “Comments to the Author” section, enter your conflict of interest statement in the “Confidential to Editor” section, and submit your "Accept" recommendation.

Reviewer #1: All comments have been addressed

Reviewer #2: All comments have been addressed

2. Is the manuscript technically sound, and do the data support the conclusions?

Reviewer #1: Yes

Reviewer #2: Yes

3. Has the statistical analysis been performed appropriately and rigorously? 

Reviewer #1: Yes

Reviewer #2: N/A

4. Have the authors made all data underlying the findings in their manuscript fully available?

Reviewer #1: Yes

Reviewer #2: Yes

5. Is the manuscript presented in an intelligible fashion and written in standard English?

Reviewer #1: Yes

Reviewer #2: Yes

6. Review Comments to the Author

Reviewer #1: Please see additional notes on the ms on the attached file (not all pages were scanned, only those containing suggested edits). A few minor corrections remain to be required for content accuracy. However, most of my previous concerns have been successfully addressed.

Reviewer #2: The revised manuscript addressed all the reviewers' suggestions and I don't have any further comments.

7. PLOS authors have the option to publish the peer review history of their article (what does this mean?). If published, this will include your full peer review and any attached files.

Reviewer #1: **Yes: **Dr. Michael Charette, PhD

Reviewer #2: No

---

## [Author Response · Author response to Decision Letter 1]

6 Jan 2025

see attached cover letter & response to reviewers. i've copied the text below, but it lost all formatting.

December 29, 2024

Dear PLOS ONE Editor,

Thank you for sharing the additional review of our manuscript (PONE-D-24-35385) entitled, “WDR75: An essential protein for ribosome assembly undergoing purifying selection.” We are very grateful for the reviewer’s attention to detail and assistance in clarifying several items in the Introduction especially. We have integrated all of their suggestions into this second revised version. Below you will find Reviewer #1’s comments followed by our responses in bold.

1. When submitting your revision, we need you to address these additional requirements. Please ensure that your manuscript meets PLOS ONE's style requirements, including those for file naming. 

Done

Reviewer #1 Comments: 

1- Some DBA is due to multiple variants in some but not all proteins.

Good catch. We clarified that DBA is due to multiple variants. The sentence in the introduction now reads, “Diamond-Blackfan anemia is caused by multiple variants in certain ribosomal proteins themselves, alternatively, diseases like Progressive Cholestasis of Northwestern Quebec (PCNQ) (formerly North American Indian childhood cirrhosis) are ribosomopathies caused by a variant in the ribosome assembly complex known as the SSU processome.”

2- SSU processome has not been introduced yet. Maybe:... a variant in the ribosome assembly complex, the SSU processome.

Great suggestion to introduce the SSU processome when first mentioning it in the introduction. The sentence now states, “...ribosomopathies caused by a variant in the ribosome assembly complex known as the SSU processome.”

3- Several annotations regarding the correction of protein names in the third paragraph of the introduction.

Thank you for the corrections. We have the following edits to maintain the accuracy of the paragraph. We clarified that WDR75 is a U3 small nucleolar RNA associated protein. We also corrected the sentence introducing t-Utp and the UtpA complex as they are different names for the same thing. The sentence now reads, “WDR75 is a key component of the transcription-Utp complex (t-Utp) which is crucial for the transcription and early processing of pre-rRNA (Gallagher, 2019).” Finally, we clarified that the protein-protein interactions that establish a binding site are WDR75, UTP4, and UTP18 instead of saying “These protein-protein interactions…”.

4- Soften as mechanism by which t-Utp regulate RNAP1 is unknown.

When discussing the essential role of WDR75 in pre-rRNA transcription, we were sure to add that the mechanism remains unknown. We qualified this part in the fourth paragraph of the introduction. It now reads, “In humans, WDR75 has multiple functions in ribosome development assembly including its essential role in pre-rRNA transcription and its role in cell survival via the p53 checkpoint. Though the mechanism by which WDR75 maintains essential subunits of RNA polymerase I is unknown, the absence of WDR75 activates RPL5/RPL11-dependent p53 stabilization checkpoint (Moudry et al., 2022).”

5- As far as I know this structure is missing many SSU processome proteins so I would not say entire and would avoid partial or incomplete as not necessary to the conversation:

You are right, thank you for noticing that detail. When introducing the 3D structure in Materials and Methods section, we have deleted the use of “entire” since the PDB does not include all the proteins in the SSU processome structure.

6- “Thoroughly analyzing the 3D structure in PyMOL, we visualized the known interacting sites with UTP4 and ECM introduced by Singh etal. (2021).” This part should be in the introduction.

Agreed. We have added the following sentence to the second paragraph of the Introduction, “The specific quaternary protein-protein interactions between WDR75, UTP4, and UTP18 establish a binding site introduced by Singh et al. (2021) at amino acids 237- 246, 288 and 321-322 in WDR75”. This helps establish that these sites were identified based on the literature.

Thank you,

Justen Whittall and Lauren Lee

jwhittall@scu.edu

(831) 332 3389

---

## [Editor Report · Decision Letter 2]

13 Jan 2025

PONE-D-24-35385R2WDR75: An essential protein for ribosome assembly undergoing purifying selectionPLOS ONE

Dear Dr. Whittall,

Thank you for submitting your manuscript to PLOS ONE. After careful consideration, we that it has merit, but I found some possible mistakes that require your further attention. I feared that you would not be able to make the changes if accepted. Therefore, I preferred to consider the option of minor changes, it will give you the opportunity to think about my comments. Please submit a revised version of the manuscript considering the corrections in case you agree with them.  SUGGESTIONS:

Linked to answer #1 and #2:

Line51. It seems there is a mistake. It is written "(*PCNQ) (formerly North American Indian childhood cirrhosis) are ribosomopathies caused by **a variant in the ribosome assembly complex known as the SSU processome**.*" Since the ribosome assembly complex known as the SSU processome is not a gene, I think the sentence should be something similar to "a variant/s in (identify the factor) some factors (if several) participating in the assembly of the first pre-ribosomal particle , or SSU processome"

By the way, line 52 contains two dots "processome. ."

Linked to answer #3:

Line 65 "*UTP4 and UTP18*",  think about including UTP6 and UTP10 as the tUTP complex seems to associate as a protein complex, composed of 7 proteins in yeast, 5 in mammals (look at our Plos one paper: Pöll, G. et al. In vitro reconstitution of yeast tUTP/UTP A and UTP B subcomplexes provides new insights into their modular architecture. PloS one 9, e114898, 2014). Do not consider the option if you think I am wrong.

Linked to answer #4:

Line 93 "*maintains essential subunits of RNA polymerase I*" I would suggest "participates in RNA polymerase I transcription" Otherwise it will be missunderstand by a direct influence on the stability of RNA polymerase I subunits, whish is not the case.

We look forward to receiving your revised manuscript.

Kind regards,

Jorge Perez-Fernandez, Ph.D.

Academic Editor

PLOS ONE
---

## [Author Response · Author response to Decision Letter 2]

14 Jan 2025

All editor suggestions integrated. See Cover Letter/Response to Reviewers statements.

---

## [Editor Report · Decision Letter 3]

16 Jan 2025

WDR75: An essential protein for ribosome assembly undergoing purifying selection

PONE-D-24-35385R3

Dear Dr. Whittall,

We’re pleased to inform you that your manuscript has been judged scientifically suitable for publication and will be formally accepted for publication once it meets all outstanding technical requirements.

Kind regards,

Jorge Perez-Fernandez, Ph.D.

Academic Editor

PLOS ONE

---

## [Editor Report · Acceptance letter]

31 Jan 2025

PONE-D-24-35385R3 

PLOS ONE

Dear Dr. Whittall, 

I'm pleased to inform you that your manuscript has been deemed suitable for publication in PLOS ONE. Congratulations! Your manuscript is now being handed over to our production team.

Kind regards, 

on behalf of

Dr. Jorge Perez-Fernandez 

Academic Editor

PLOS ONE